# Heart Failure and Major Adverse Cardiovascular Events in Atrial Fibrillation Patients: A Retrospective Primary Care Cohort Study

**DOI:** 10.3390/biomedicines11071825

**Published:** 2023-06-26

**Authors:** P. Moltó-Balado, S. Reverté-Villarroya, C. Monclús-Arasa, M. T. Balado-Albiol, S. Baset-Martínez, J. Carot-Domenech, J. L. Clua-Espuny

**Affiliations:** 1Primary Health-Care Centre Tortosa Oest, Institute Català de la Salut, Primary Care Service (SAP) Terres de l’Ebre, 43500 Tortosa, Spain; pemolto@gmail.com (P.M.-B.); cmonclusa.ebre.ics@gencat.cat (C.M.-A.); 2Biomedicine Doctoral Programme, Universitat Rovira I Virgili, 43500 Tortosa, Spain; 3Biomedicine Doctoral Programme, Advanced Nursing Research Group, Nursing Department, Campus Terres de l’Ebre, Universitat Rovira I Virgili, 43500 Tortosa, Spain; 4Conselleria de Sanitat, Departament de Salut de La Plana, Primary Health-Care Centre CS Burriana I, 12540 Burriana, Spain; balado_maralb@gva.es; 5Nusing Management Primary Health-Care Center Tortosa Est, Institut Català de la Salut, Primary Care Service (SAP) Terres de l’Ebre, 43500 Tortosa, Spain; sbaset.ebre.ics@gencat.cat; 6Direction of Information and Communication Systems, Territorial Management of Terres de l’Ebre, Institut Català de la Salut, 43500 Tortosa, Spain; jcarot.ebre.ics@gencat.cat; 7Ebrictus Research Group, Research Support Unit Terres de l’Ebre, Institut Universitari d’Investigació en Atenció Primària Jordi Gol (IDIAP Jordi Gol), 43500 Tortosa, Spain; 8Primary Health-Care Center Tortosa Est, Institut Català de la Salut, Primary Care Service (SAP) Terres de l’Ebre, 43500 Toosa, Spain

**Keywords:** atrial fibrillation, risk-atrial fibrillation, heart failure, outcomes, major adverse cardiovascular events (MACE), major outcomes, vascular events, primary health care

## Abstract

Background: Atrial fibrillation (AF) is a common cardiac arrhythmia that is associated with an increased risk of major adverse cardiovascular events (MACE). The main goal was to analyze the links and associations between AF and MACE. Methods: A multicenter, observational, retrospective, community-based study of a cohort (*n* = 40,297) of the general population aged 65–95 years between 1 January 2015 and 31 December 2021 without a previous diagnosis of AF or MACE in the Primary Care setting. Results: 2574 people (6.39%) developed a first AF event, resulting in an overall incidence of 8.9/1000 people-years [CI95% 8.6–9.2]. The incidence of MACE among those with AF was 75.1/1000 people-years [CI95% 70.8–79.5], whereas among those without AF, it was 20.6/1000 people-years [CI 95% 20.2–21.1], resulting in a rate ratio of 3.65 [CI 95% 3.43–3.88, *p* < 0.001]. Besides, the incidence of HF with AF was 40.1 people-years [CI 95% 37.1–43.2], while in the group without AF, it was 8.3 people-years [CI 95% 7.9–8.6, *p* < 0.001], with a rate ratio of 4.85 [CI 95% 4.45–55.3, *p* < 0.001]. Before an AF diagnosis, there is already a higher risk of chronic kidney disease, ischemic cardiopathy, and peripheral artery disease. A higher risk of poor nutritional status was detected among those with MACE (49.7% vs. 26.6%, *p* < 0.001). Conclusions: AF diagnosis increases the incidence of heart failure fourfold. Additional information is required to establish the connection between AF, major adverse cardiovascular events, and nutritional status.

## 1. Introduction

Major adverse cardiovascular events (MACE) [1] are known as the composite of total death, myocardial infarction, coronary revascularization, stroke, and heart failure (HF). These events contribute to significant all-cause morbidity and mortality, decreased quality of life, and increased medical costs [2,3]. Heart failure (HF) is a growing global problem that is expected to increase in the coming years due to population aging, an increase in cardiovascular risk factors, and improvements in the management of acute cardiovascular events. On the one hand, it is the leading cause of hospitalization in patients over 65 years old and the third leading cause of cardiovascular mortality. On the other hand, HF can be caused by a variety of factors, including AF, which is a leading cause of morbidity and mortality, with an estimated five million incident cases globally. It is estimated that the prevalence of AF will increase from 1.9% (2008) to 3.5% (2050), and the number of AF-related ischemic strokes in people >80 years will triple (2010–2060) [4] in both developing and developed countries. AF occurs in more than half of individuals with HF, and HF occurs in more than one-third of individuals with AF [5]. The potential association between HF and AF makes them one of the chronic conditions with the greatest health and economic impact [6].

Other factors have been associated with an increased risk of MACE in patients with AF [1,2,3,7,8] as well as hypercholesterolemia [9], malnutrition [10,11], or an elevated CHA2DS2-VASc score [12,13], a commonly increased score in AF patients, but both HF and AF, along with factors associated with MACE, are frequently underdiagnosed. In the last three decades, new therapeutic targets have allowed for the modification of the natural history of heart failure, but mortality rates and recurrent hospitalizations remain very high in patients with HF, suggesting that additional measures are needed. Identifying high-risk populations for AF and detecting it early can help reduce the burden of MACE associated with heart failure and reduce their risk of serious complications. The European Action Plan (2018–2030) [14] considers the availability of screening and treatment programs for stroke risk factors in Europe to be important.

Traditional practice advocates the use of clinical risk score criteria to distinguish at-risk patients, but these risk scores have modest discriminatory power. New insights have been gained into the usefulness of biomarkers and imaging techniques, and data are emerging on the importance of subclinical device detection using portable devices to recognize cardiac arrhythmias in primary care practice. The main goal of this study was to compare the characteristics of patients developing their first episode of diagnosed AF during the follow-up period and analyze links and associations between AF and MACE.

## 2. Materials and Methods

### 2.1. Study Design

The study analyzes links and associations between AF and MACE among 40,297 people from the general population aged between 65 and 95 years old residing in the region of Ebre’s lands, Catalonia, Spain. It was an observational, retrospective, and community-based study conducted between 1 January 2015, and 31 December 2021.

### 2.2. Study Scope

The study was carried out in Terres de l’Ebre (Health Region Terres de l’Ebre, Appendix A), located in the southern part of Catalonia (Spain).

The territory is made up of 4 counties with 11 primary care teams (EAPs), managed by the Catalan Health Institute (ICS), Department of Health (CatSalut). Specialized care is received at the reference hospital located in Tortosa, “Hospital Verge de la Cinta”, which is publicly managed by the ICS. The EAPs are organized as independent clinical functional teams. The majority of the census population in the territory (98.2%) has an active Shared Health Record of Catalonia (HC3) available digitally for continuous care monitoring from any center.

### 2.3. Data Collection and Information Sources

Data for all participants were managed by ICS through the 11 EAPs. The Department of Information and Communication Technologies, through the registration of the minimum basic dataset at hospital discharge (CMBD-HA), retroactively provided an anonymized computerized database with the clinical history using the specific International Classification of Diseases (10th version; ICD-10) to the principal investigator in a fully de-identified format. 

The data sets utilized for this project were as follows: The Institute of Statistics of Catalonia collected data on aging index, inhabitant density, and gross disposable household income per inhabitant for each region in Catalonia [14,15,16,17].The Health Plan of the Terres de l’Ebre Healthcare Region 2021–2025 [14] outlines healthcare goals, priorities, and actions for the region.The HC3 Patient Episode Dataset contains demographic and clinical information on inpatient and outpatient care in Catalan hospitals.The 11 EAPs managed by the Catalonian Health Institute share a clinical information database for general practice and hospital interactions, including clinical data, diagnoses, medication, referrals, and patient status as of 31 December 2021.

### 2.4. Ethical Aspects and Data Protection

The data were analyzed and supervised according to the General Data Protection Regulation of Spain/Europe from 1 February 2017. The study was conducted in accordance with the most relevant standards regarding data handling, concerning the experimental context with patients, ethics, and data protection and privacy, following Directive 95/46/EC (protection of individuals with regard to the processing of personal data and on the free movement of such data). All of the data were included in an ad hoc repository, which was delivered to the main researcher. The study protocol received ethics evaluation and approval from the Ethical Committee of Jordi Gol University Institute of Primary Care Research with registration number 22/243-P.

### 2.5. Study Population 

Initially, the study included individuals over 65 years old (*n* = 55,459) who did not have a history of AF or major adverse cardiovascular events in their medical records. The following criteria were defined:Outcomes: The new diagnosis of AF was the primary outcome. Secondary outcomes were major adverse cardiovascular events, cognitive impairment, and all-cause mortality.Inclusion criteria: Subjects 65–95 years of age who met the inclusion criteria: high-risk AF (according to the risk model and belonging to Q4) [18,19], active medical history in any of the health centers in the territory with information accessible through the shared history (HC3), without prior AF, residence in the territory, and assignment to any of the Primary Care Teams (EAP) of the same.Exclusion criteria: Persons under 65 or over 95 years of age; population who are not from Terres de l’Ebre; patients without a previous diagnosis of AF; treatment with anticoagulants; impairment of cognitive status; Barthel score <55 points; pacemaker or defibrillator wearer. The non-availability or loss of accessibility to the information necessary for the study was considered a reason for exclusion.

After excluding patients because they did not fit the inclusion criteria or due to a lack of the appropriate variables to categorize the risk of AF, 40,297 people (Figure 1) joined the trial. All participants were monitored from the date of inclusion (1 January 2015) until 31 December 2021 loss-to-follow-up, or date of death, whichever happened first.

### 2.6. Variables

The data on AF and comorbidities for cardiovascular risk trajectories lasted until loss-to-follow-up, date of death, or 31 December 2021, whichever happened first. AF was diagnosed according to the guidelines of the European Society of Cardiology [19]. The diagnosis of AF was verified by two investigating physicians blinded to the diagnosis. When a consensus was not reached, a cardiologist was consulted. Patients were classified according to the presence of AF. In the case of AF diagnosed during the follow-up period, data were collected at the time of AF diagnosis or until the end of the follow-up. MACE after AF diagnosis until the end of follow-up were analyzed. MACE prior to AF diagnosis were not analyzed. Data for patients who did not present with AF during follow-up were obtained in the last year of follow-up.

Sociodemographic: age, sex, primary care team, and region.Cardiovascular risk factors and diagnostics using specific ICD–10 code prefixes for hypertension (I10–I15), hypercholesterolemia (E78), smoking (F17.203, Z72), body mass index (BMI), diabetes mellitus (E10–E14), sleep apnea-hypopnea syndrome (G47.3), heart failure (I50-51), ischemic heart disease (myocardial infarction, percutaneous coronary intervention, stable or unstable angina or coronary artery bypass grafting) (I20–I25), chronic kidney disease (CKD) (N18), and estimated glomerular filtration rate (eGFR mL/min/1.73 m^2^), cerebrovascular illness (transient ischemic attack or ischemic stroke) (I63, G45), COPD, asthma, chronic bronchitis (J40–J45), cancer (C00–C96), and COVID-19 (U07.1). Coronary artery disease was defined as either a history of myocardial infarction, coronary bypass graft surgery, and/or percutaneous transluminal coronary angioplasty. Data on COVID-19 was collected from 15 March 2020 (first wave in Spain) to 31 December 2021.Clinical scores: risk-index AF [18,19], stroke risk by CHA_2_DS_2_-VASc score, Barthel Index for Activities of Daily Living (ADL), controlling nutritional status (CONUT) score, and Adjusted Morbidity Groups (GMA) score as recommended by current guidelines [20]. The model to categorize the risk of suffering AF at five years among community members ≥65 years old was previously published [18,19]. It includes the following variables: sex, age, average heart rate, average weight, and CHA_2_DS_2_VASc score. The mathematical formula of the model was applied to the target population without a diagnosis of AF, and the quartiles of the distribution from lowest to highest risk were defined (Q1–Q4), with Q4 (high risk) being of interest, though the AF incidence density/1000 people-years (ID) was calculated for each group, as was the incidence of MACEs and the registered prevalence of cognitive decline.Pharmacological treatment: antiplatelet agents, new anticoagulants, and antivitamin K.Final status: dead/alive.

### 2.7. Statistical Analysis

The characteristics of the population were defined through a descriptive statistical analysis. Baseline data are presented as numbers and percentages, mean and standard deviation (SD), or median and interquartile range (IQR), as appropriate. Qualitative variables were analyzed with the chi-square distribution according to bivariate analysis for normal distributions, while quantitative variables were examined with Student’s t-distribution for independent samples. The calculation of the incidence rate took into account the total observation time for each person in the population, reflecting the duration of both disease risk and monitoring. 

To assess the increased risk of vascular outcomes associated with AF, hazard ratios are calculated using Cox proportional hazards regression analysis. The hazard ratios were adjusted by including the significant confounding variables in the regression model. Any variables that were found to have a significant *p*-value (≤0.05) and were not included in the scores used (CONUT and CHA_2_DS_2_-VASc) were considered potential confounding factors. Absolute risk increases were reported in terms of events per 1000 person-years of follow-up. Cox regression was utilized to compare MACE incidence between the AF and non-AF groups, while Kaplan–Meier curves were used to evaluate mortality. IBM SPSS Statistics version 21.0 was utilized for statistical analysis and data management. 

## 3. Results

### 3.1. Baseline Characteristics

Patient baseline characteristics are shown in Table 1. A total of 40,297 people without a personal history of AF were included. The average follow-up time was 80.65 ± 9.5 months. The study population had an average age of 77.65 ± 8.46 years, with 46.48% being women who were significantly older than men (81.22 ± 7.91 vs. 77.65 ± 8.46 years, *p* < 0.001). 

### 3.2. AF Incidence

A total of 2574 people (6.39%) developed a first AF event after a median follow-up time of 78.6 ± 12.1 months. The overall incidence was 8.9/1000 people-years [CI 95% 8.6–9.2], significantly higher among men [9.8/1000 people-years, CI95% 9.3–10.3 vs. 8.1/1000 people-years, CI95% 7.7–8.5; *p* < 0.001]. There were significant differences between the AF patterns for all risk factors of interest at baseline (Table 1). The average age of 81.2 ± 7.9 was significantly higher than the average age of the overall study population. 

Participants with AF had a significantly higher prevalence of major adverse cardiovascular events (MACE) compared to those without AF (40.7% vs. 12.7%, *p* < 0.001). Additionally, they had higher average scores (*p* < 0.001) on the CHA_2_DS_2_-VASc and CONUT scales, as well as higher overall mortality rates (18.02% vs. 20.12%, *p* < 0.001).

### 3.3. MACE Incidence among AF vs. No-AF Patients

During 15,779 patient-years of follow-up, we observed a total of 2574 AF new diagnoses and 1748 episodes of MACE (Table 2). The overall incidence rate of MACE in the group with AF was 73.0/1000 people-years [CI95% 68.9–77.1], while in the group without AF, it was 21.1/1000 people-years [IC95% 20.5–21.6, *p* < 0.001], with a rate ratio of 3.52 [CI95% 3.31–3.75, *p* < 0.001]. The overall incidence rate of HF in the group with AF was 40.1 people-years [CI95% 37.1–43.2], while in the group without AF, it was 8.3 people-years [IC95% 7.9–8.6, *p* < 0.001], with a rate ratio of 4.85 [CI95% 4.45–55.3, *p* < 0.001] (Figure 2).

In this comprehensive review of the association between AF and the risk of MACE, various results were observed (Table 2), including a 3.65-fold increased risk of a major cardiovascular event, a 2.38-fold increased risk of ischemic heart disease, a 4.97-fold increased risk of congestive heart failure, a 5.04-fold increased risk of ischemic stroke, and a 2.57-fold increased risk of all-cause mortality.

Eventually, among patients diagnosed with AF, 718 (27.9%) were not treated with anticoagulants (NOAC or vitamin K antagonist). A total of 392 (25.7%) in the AF+MACE- group and 326 (31.1%) in the AF+MACE+ group were on anticoagulant treatment. On the other hand, out of the 1220 (2.3%) patients without a diagnosis of AF, 899 (1.7%) in the AF-MACE- group and 321 (0.6%) in the AF-MACE+ group were treated with anticoagulants.

As the risk of AF increased from Q1 to Q4, there was a corresponding increase in the incidence of MACE. In fact, being in Q4 doubled the risk of most vascular events included in the study (Table 2). The results showed a significantly increased risk (*p* < 0.001) of heart failure, stroke, and MACE among individuals newly diagnosed with AF. However, there were no significant differences in the incidence of peripheral artery disease (*p* = 0.724), ischemic cardiomyopathy (*p* = 0.908), or chronic kidney disease (*p* = 0.706). Furthermore, AF was found to be associated with a 1.34-fold increased risk of cognitive impairment (*p* < 0.001), and the overall mortality rate was higher among individuals in the fourth quartile.

### 3.4. Nutritional Status Assessed by CONUT Score 

Using the CONUT scale (Table 3), 49.7% of patients with AF were detected to have malnutrition (*p* < 0.001). The presence of MACE doubles the risk of malnutrition (*p* < 0.001). 

### 3.5. Regression Cox Model

After adjusting for age, genre, BMI, cardiovascular risk factors, antiaggregants, and anticoagulants, only the CHA_2_DS_2_-VASc, Charlson score, and CONUT scores were retained as independent prognostic factors (Table 4) for major adverse cardiovascular events among individuals with a new diagnosis of AF.

## 4. Discussion

In this large study, the incidence of AF was associated with a higher incidence of MACE (heart failure, ischemic heart disease, stroke, and mortality), CKD, and cognitive impairment. The relative and absolute risk increase associated with many of these events is higher than that of those subjects without AF. Several studies have reported similar findings regarding the increased risk of most major cardiovascular events in individuals diagnosed with AF [1,5,6]. Specifically, AF has been associated with a higher prevalence of heart failure, particularly in older adults and those with pre-existing cardiovascular disease. Moreover, the study findings highlight that the risk of cardiovascular events is already elevated prior to the diagnosis of AF, particularly in patients with CKD, ischemic cardiopathy, and peripheral artery disease who are classified as high-risk for developing AF. 

AF is not only frequently undiagnosed but also commonly left untreated and unmanaged [21,22,23,24]. Moreover, AF and heart failure often coexist and can worsen each other’s impact. Anticoagulation therapy has been shown to improve outcomes in patients with HF and AF, particularly by reducing the risk of stroke and other thromboembolic events, which can cause further damage to the heart and worsen HF [25,26]. Additionally, some studies have suggested that anticoagulation therapy may have direct anti-inflammatory effects on the heart, which can help improve heart function and reduce HF symptoms [27]. The increase in risk for most major cardiovascular events associated with a diagnosis of AF is higher than those described in other studies [6,8]. In general, the risk of MACE in patients with AF is estimated to be approximately double that of those without AF, and its magnitude varies according to the studied population, individual risk factors, and the control of risk factors, especially the indication of anticoagulant treatment and, in the case of using a vitamin K antagonist, achieving the goals within the therapeutic range. These indicators were shown to be deficient in previous research [28].

Moreover, the undiagnosed and poor control of cardiovascular risk factors associated with AF [1] is known from common evidence, and its incidence may be higher than previously thought. It is uncertain how exactly AF contributes to an increased risk of various cardiovascular diseases, but it is possible that AF may serve as an indicator of common underlying risk factors for cardiovascular disease. These risk factors may include hypertension, which is present in as many as 90% of AF patients, as well as obesity, diabetes, obstructive sleep apnea [29], and the CHA_2_DS_2_-VASc score. However, it is important to note that AF is a treatable condition, and effective management of AF can help reduce the risk of heart failure and mortality [8]. Close monitoring and management of underlying cardiovascular risk factors can also be helpful in reducing the risk of complications associated with AF.

The CHA_2_DS_2_-VASc score is a widely used tool for assessing the risk of stroke in patients with AF. It takes into account several risk factors that are associated with both AF and heart failure, such as age, hypertension, diabetes, and previous cardiovascular disease. A higher CHA_2_DS_2_-VASc score is associated with an increased risk of developing heart failure and an increased risk of MACCEs [29] in patients with AF [30]. Therefore, the CHA_2_DS_2_-VASc score can be a useful tool for identifying patients with AF who are at high risk of developing heart failure and who may benefit from more aggressive management of their cardiovascular risk factors. It may also have beneficial effects on heart failure outcomes.

Women are often older at the time of diagnosis and have a higher prevalence of hypertension and valvular heart disease [31], and AF is a stronger risk factor for cardiovascular disease and death in women compared with men [32]. Although decisive evidence is pending, it is suggested that the structural development of AF differs, with women tending to have more atrial fibrosis and different patterns of electrical function. This may imply differences in the underlying pathophysiology between men and women. In this study, women had a higher prevalence of AF and a higher age average at diagnosis than men, according to current evidence [21,32]. Perhaps the subgroup of women in Q4 should be considered for both AF screening and close monitoring of modifiable risk factors, given their higher incidence of comorbidities, stroke, and severe disability [14,19,32]. Consequently, clinicians should be aware of the importance of detecting and providing appropriate therapies for MACE prophylaxis, as well as the downstream economic burden on an increasingly aging population with an increased incidence of AF [6,21]. 

Several nutritional alterations have been described and underdiagnosed in AF patients [33], but little is known about the nutritional status of AF patients and the relationship between malnutrition and mid- and long-term mortality. The study highlighted that patients with AF and MACE had a higher prevalence of some degree of malnutrition, which is consistent with the results of previous studies [10,34,35,36]. This fact may reflect the social needs and social determinants of health (SDOH) leading to poorer health outcomes [37,38]. Therefore, healthcare providers should address social needs and SDOH in their patient care to reduce prevalent healthcare disparities. Further information is needed to determine the relationship between AF, MACE, and nutritional status. Likewise, the detection of nutritional alterations in individuals with cardiovascular risk should be evaluated as a target subgroup for AF screening and MACE.

The statement highlights an important finding from a study that suggests that certain conditions such as chronic kidney disease, ischemic cardiopathy, and peripheral artery disease may increase the risk of developing AF and heart failure. The study further suggests that this risk may be present even before the diagnosis of AF is made, especially among individuals in the fourth quartile. Subclinical AF has been associated with an increased risk of stroke [39], but there is limited understanding of their temporal relationship. Individuals with chronic kidney disease are at a higher risk of developing AF and heart failure. This may be due to the fact that chronic kidney disease can lead to changes in the structure and function of the heart, which can increase the likelihood of developing AF and HF [38]. Similarly, ischemic cardiopathy and peripheral artery disease can also lead to changes in the heart that increase the risk of developing AF and MACE [29,40]. These conditions can lead to changes in the structure and function of the heart, which could also increase the likelihood of developing AF [41,42] and emphasize the importance of managing these risk factors in patients at risk of AF. However, it should be noted that individuals with prevalent AF are more likely to develop these conditions. Therefore, more aggressive monitoring and treatment of this high-risk population may improve outcomes. 

Additionally, it should be pointed out that there was a significant increase in the incidence of cognitive impairment and mortality from the 4th quartile group (Q4) to the group with newly diagnosed AF. A high incidence of silent cerebral infarction detected by MRI and AF-induced cognitive dysfunction related to silent cerebral infarction has been reported [43]. It has been suggested that AF ablation may also reduce the risk of MACE in selected patients [44], such as those with heart failure and left ventricular dysfunction. However, further studies are needed to confirm these findings and determine which patients may benefit most from AF ablation in terms of reducing the risk of MACE.

The prevalence of sleep apnea recorded in the study was 4.9%, which is higher than the published prevalence [45,46]. It would be interesting to examine the characteristics of AF patients with a high probability of sleep apnea, such as males, smokers, and individuals with an increased BMI [47]. Additionally, the interaction between clinical risk factors remains uncertain, and several combinations of risk factors may carry a higher risk when examined together compared to other combinations. Additionally, the role of AF burden in patients with subclinical paroxysmal AF is still under debate [48].

The Charlson comorbidity index is a tool used to measure the severity and impact of multiple comorbid conditions on a patient’s health status. Both AF and heart failure are common comorbidities that may increase a patient’s Charlson score, which was associated with a higher risk of developing AF independent of other risk factors [49], a greater risk of mortality in patients with HF [50], and an increased risk of hospitalization and mortality [51]. Emphasizing the impact of the association between AF and cardiovascular comorbidities across a wide spectrum raises an important point about the “chicken and egg” relationship between AF and cardiovascular disease. Its temporal relationships have not yet been fully explored, though it has been described as having a lower risk for ischemic stroke in prevalent HF than in incident HF and higher mortality and a higher risk of re-hospitalization for HF among patients in whom HF preceded AF [52]. This approach offers several advantages, including a more comprehensive understanding of the disease, facilitation of holistic patient care, improvement in risk assessment, enablement of targeted interventions, and promotion of further advancements in research and innovation.

Given that the COVID-19 pandemic occurred during the study period and contracting the disease was associated [53,54] with an increased risk of thromboembolism and a higher risk of mortality, it was included as a baseline variable. Several studies [55,56,57,58] have suggested that there may be a relationship between COVID-19 and an increased risk of developing MACE, HF, and AF, particularly in patients with pre-existing cardiovascular disease. Given that the exact mechanisms by which COVID-19 may increase the risk of MACE, heart failure, and AF are not fully understood, more evidence from ongoing clinical studies is necessary to identify possible criteria for developing MACE and the impact of thromboprophylaxis, as well as outcome factors on MACEs. 

One of the key strengths of this study is the large sample size because several populations require special assessment when considering individualized risk stratification, and these populations are poorly represented in the original derivation cohorts for clinical risk scores; as such, the applicability of such scores is limited. On the other hand, some potential limitations need to be taken into account in the interpretation of our results. Thromboembolic and AF risk scores are highly effective in determining the risk of the population, but they are often misleading when applied to individuals. This is especially true for low-risk patients. Due to the observational study design, we are not able to prove causality, and residual confounding may persist despite comprehensive multivariable adjustment.

The project will continue by utilizing a matching learning methodology (artificial intelligence) to discover patterns and correlations between various variables and develop predictive models. It may be the case that some variables with the potential to impact outcomes have not been recorded, and there is a chance that little details may not be accurately recorded. The authors used registered cohorts in all of their analyses, which is an approach that helps to overcome comparability limitations that arise due to the heterogeneity of the available data. However, it should be noted that the use of the registration system and territorial organization as the basis for this approach can be considered a common limitation. The study design does not allow for an answer to this limitation.

## 5. Conclusions

Patients with atrial fibrillation have a significantly higher incidence of heart failure, with a four-fold increase in risk. Additionally, both the CHA_2_DS_2_-VASc, Charlson, and CONUT scores have been identified as independent prognostic factors for MACE-related AF. The risk of developing heart failure is already elevated prior to AF diagnosis, especially in patients with chronic kidney disease, ischemic cardiopathy, and peripheral artery disease in the fourth quartile. To identify modifiable predictors of MACE, it may be useful to explore various tools for detecting AF and implement preventive interventions in primary care.

## Figures and Tables

**Figure 1 biomedicines-11-01825-f001:**
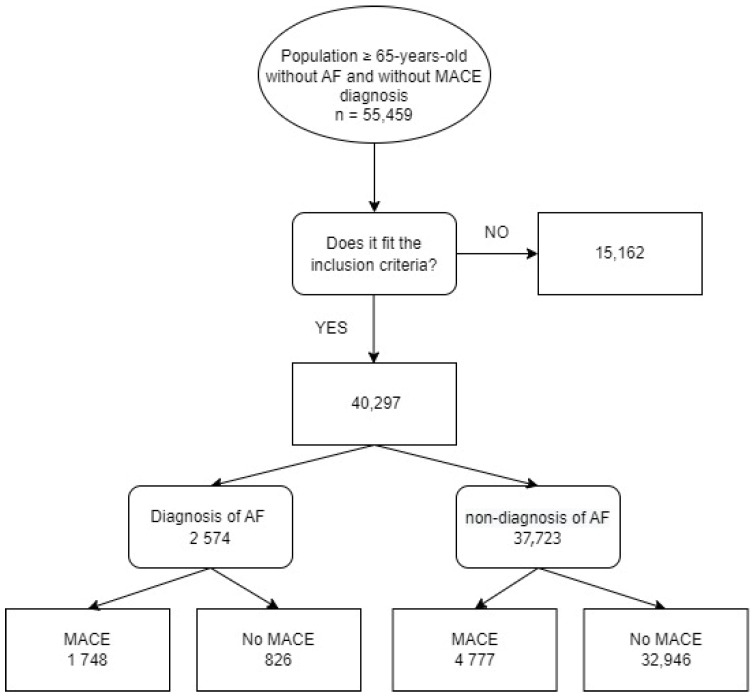
Flowchart of the selection process according to the inclusion and exclusion criteria. MACE: major adverse cardiovascular events; AF: atrial fibrillation.

**Figure 2 biomedicines-11-01825-f002:**
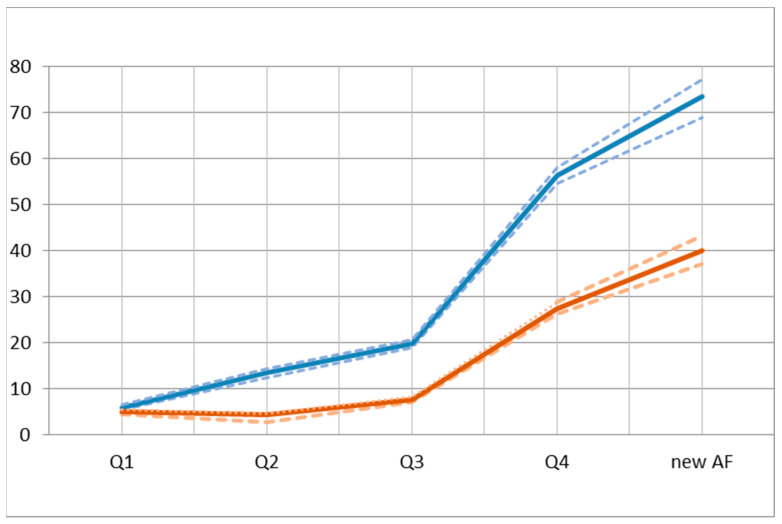
Major adverse cardiovascular events (MACE) and heart failure (HF) incidence density rates for every 1000 people-years (IC95%) by risk quartiles vs. new atrial fibrillation. The thick line indicates the CI and the dashed lines show the confidence intervals.

**Table 1 biomedicines-11-01825-t001:** Baseline characteristics of cases with AF vs. without AF.

Variables	No AF	(%)	AF	(%)	*p*	ALL
All (*n* %)	37,723	93.6%	2,574	6.4%	-	40,297
Women	17,535	46.5%	1343	3.3%	<0.001	18,878
Age average	77.6 ± 8.7		81.2 ± 7.9		<0.001	77.9 ± 8.5
CHA_2_DS_2_-VASc	3.2 ± 1.1		3.8 ± 1.2		<0.001	3.2 ± 1.2
Hypertension arterial	23,610	62.6%	1945	75.6%	<0.001	25,555
Diabetes mellitus	9689	25.7%	769	30%	<0.001	10,458
Dyslipidemia	17,913	47.5%	1216	47.3%	0.822	19,129
BMI ^1^ (kg/m^2^)	28.7 ± 5.1		29.5 ± 5.4		<0.001	28.7 ± 5.2
Ischemic cardiomyopathy	2558	6.8%	357	13.9%	<0.001	2915
Heart failure	2096	5.6%	676	26.·%	<0.001	2772
Stroke/TIA	698	1.9%	187	7.3%	<0.001	885
Vascular peripheral disease	2431	6.4%	345	13.4%	<0.001	2776
Dementia/cognitive impairment	3471	9.2%	310	12.1%	<0.001	3781
Liver disease	72	0.2%	10	0.4%	0.04	82
Chronic kidney disease	5158	13.7%	676	26.3%	<0.001	5834
Glomerular filtration rate (mL/min/1.73 m^2^)	72.9 ± 18.6		63.5 ± 20.4		<0.001	72.2 ± 19
Thyroid disease	2613	6.9%	215	8.3%	0.047	2828
OSAHS ^2^	1022	2.7%	126	4.9%	<0.001	1148
COPD ^3^/asthma/bronchitis	4591	12.2%	447	17.4%	<0.001	5038
CONUT	0.8 ± 1.3		1.3 ± 1.5		<0.001	0.8 ± 1.3
Serum albumin (g/dL)	5.5 ± 10.5		5 ± 10.7		0.029	5.5 ± 10.4
Lymphocytes (×10^3^/μL)	2.4 ± 17.5		2.1 ± 1.3		0.312	2.4 ± 16.8
Statins	11,806	31.3%	945	36.7%	<0.001	12,751
Antiaggregants	6110	16.2%	141	5.5%	<0.001	6251
Anticoagulation	987	2.6%	1994	77.5%	<0.001	2981
VKA ^4^	754	2%	944	36.7%	<0.001	1698
NOAC ^5^	235	0.6%	1053	40.9%	<0.001	1288
CHARLSON	1.3 ± 1.3		1.8 ± 1.4		<0.001	1.3 ± 1.3
Average follow-up time	80.8 ± 9.3		78.6 ± 12.1		<0.001	80.7 ± 9.5
COVID-19	2931	7.8%	260	10.1%	<0.001	3191

^1^ BMI: body mass index; ^2^ OSAHS: obstructive sleep apnea-hypopnea syndrome; ^3^ COPD: chronic obstructive pulmonary disease; ^4^ VKA: vitamin K antagonist; ^5^ NOAC: non-vitamin K antagonist oral anticoagulant.

**Table 2 biomedicines-11-01825-t002:** Association between AF diagnosis and MACE according to risk quartile.

	Q4	No-AF	New AF	HR AF/Q4	HR AF/No-AF
N	10,239	37,723	2574		
Age (average ± SD)	84.8 ± 6.7	77.65 ± 8.4	81.2 ± 7.9		
AF (n) Incidence/1000 people-years [CI95%]	1148 17 [16.1–18.1]	-	2574 8.9 [8.6–9.2]		
Chronic kidney disease (n %) Incidence/1000 people-years [CI95%]	2748 (26.83%) 40.8 [39.3–42.3]	5158 (13.67%) 20.3 [19.8–20.9]	676 (26.26%) 40.1 [37.1–43.2]	0.98 [0.90–1.06] *p* = 0.706	1.97 [1.82–2.13] *p* < 0.001
Cognitive impairment (n %) Incidence/1000 people-years [CI95%]	1569 (15.32%) 23.3 [22.1–24.5]	3471 (9.2%) 13.7 [13.2–14.1]	310 (12.04%) 18.4 [16.4–20.6]	0.78 [0.69–0.89] *p* = 0.002	1.34 [1.2–1.51] *p* < 0.001
Heart failure (n %) Incidence/1000 people-years [CI95%]	1853 (18.1%) 27.5 [26.3–28.8]	2096 (5.56%) 8.3 [7.9–8.6]	676 (26.26%) 40.1 [37.1–43.2]	1.45 [1.33–1.6] *p* < 0.0001	4.85 [4.5–5.3] *p* < 0.0001
Ischemic heart disease (n %) Incidence/1000 people-years [CI95%]	1479 (14.44%) 22.0 [20.8–23.1]	2558 (6.78%) 10.1 [9.7–10.5]	367 (14.26%) 21.8 [19.6–24.1]	0.99 [0.88–1.11] *p* = 0.908	2.16 [1.93–2.41] *p* < 0.001
Stroke/transient ischemic attack (n %) Incidence/1000 people-years [CI95%]	459 (4.48%) 6.8 [6.2–7.5]	698 (1.85%) 2.7 [2.5–3.0]	187 (7.26%) 11.1 [9.6–12.8]	1.62 [1.37–1.92] *p* < 0.001	4.03 [3.43–4.74] *p* < 0.001
Peripheral arteriopathy (n %) Incidence/1000 people-years [CI95%]	1347 (13.15%) 20.0 [18.9–21.1]	2431 (6.44%) 9.6 [9.2–10.0]	345 (13.4%) 20.5 [18.4–22.7]	1.02 [0.90–1.15] *p* = 0.724	2.13 [1.90–2.4] *p* < 0.001
Death (n %) Incidence/1000 people-years [CI95%]	2861 (27.94%) 42.5 [40.9–44.0]	6799 (18.02%) 26.8 [26.1–27.4]	518 (20.12%) 30.7 [28.1–33.5]	0.72 [0.65–0.79] *p* < 0.001	1.14 [1.04–1.25] *p* = 0.027
Total MACE (n%) Incidence/1000 people-years [CI95%]	3791 (37.02%) 56.3 [54.5–58.1]	5352 (14.11%) 21.1 [20.5–21.6]	1748 (67.9%) 73.0 [68.9–77.1]	1.29 [1.21–1.38] *p* < 0.001	3.52 [3.31–3.75] *p* < 0.001

**Table 3 biomedicines-11-01825-t003:** Nutritional status and prevalent AF with MACE.

CONUT Risk (n)	[AF + MACE+] 1748	[AF+ MACE−] 826
Normal (1–2)	879 (50.3%)	606 (73.4%)
Light (2–4)	671 (38.4%)	187 (22.7%)
Moderate (5–8)	137 (7.8%)	31 (3.7%)
Severe (9–12)	61 (3.5%)	2 (0.2%)

**Table 4 biomedicines-11-01825-t004:** Prognostic independent factors of MACE incidence among AF people.

Variables	Hazard Ratio	CI95%	*p*
CHA_2_DS_2_-VASc score	2.50	2.41–2.57	<0.001
CONUT score	1.06	1.04–1.08	<0.001
Charlson score	1.24	1.21–1.27	<0.001

## Data Availability

The data that support the findings of this study are available from the corresponding author (S.R.-V. and J.L.C.-E.) upon reasonable request.

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
