# Peer review of "Heart Failure and Major Adverse Cardiovascular Events in Atrial Fibrillation Patients: A Retrospective Primary Care Cohort Study"

_biomedicines, 2023, doi:10.3390/biomedicines11071825_

Round 1

Reviewer 1 Report

The authors try to find ties between AF and MACE in a large population of 40.000 patients in rural spain.

The ties between AF, CKD, HF and other comorbidities have already been described as shown in the introduction.

As a retrospective observational trial it has many limitations which i m sure the authors already know e.g. confounding factors : patients where included if they had a medical history in one of the centers, meaning those where likely sicker patients to begin with.

The english are well written and the figures are nicely prepared.

Author Response

The authors are very grateful for a new chance of considering the manuscript and have been keen to answer the required comments and suggestions. All modifications can be seen in red in the manuscript.

Reviewer 2 Report

The main goal of this study was to compare characteristics of patients developing first episode of diagnosed atrial fibrillation during follow-up period, and analyze links and associations between AF and MACE. The manuscript is difficult to go through .

Comments /Queries

1. In the introduction it is stated "Major adverse cardiovascular events (MACE) are known as the composite of total death, myocardial infarction, coronary revascularization, stroke, and hospitalization because of heart failure (HF)", whereas in the beginning of the discussion " In this large study, the incidence of AF is associated with a higher incidence of MACEs including major cardiovascular events, heart failure, ischemic heart disease, stroke, CKD, cognitive impairment and all-cause mortality" What is MACE?

2. Materials and methods section. Confusing, difficult to go through, and unnecessarily long. It should be halved, abbreviations should be kept to a minimum and be much more clear.

3. Figure 1. It is very confusing and should be replaced. At the beginning one understands that the study included patients without MACE and AF but in the end one sees two group of patients with MACE without and with atrial fibrillation.

4. When analyzing the data the authors should keep in mind the "chicken and egg" relationship between atrial fibrillation and cardiovascular disease. This, is very important.

Minor editing required.

Author Response

(The authors gave the same response as above.)

Reviewer 3 Report

This review article by Molto-Balado P et al. focused on the association between atrial fibrillation (AF) and major adverse cardiovascular events (MACE). Authors evaluated the incidence of AF first, and observed the incidence of MACE. Then, athors demonstrated that AF was significantly associated with an increased risk of MACE. As authors mentioned, AF was reportedly associated with heart failure, and vice versa. However, the investigations of the influence of AF on other MACE remain limited. Therefore, the concept of this study to clarify it is understandable and results seem agreeable. Although the concept seems good, authors may want to consider server issued as follows.

Major comment

1) In Abstract, although authors mentioned the incidence of heart failure in conclusion, authors did not show the results about hear failure anywhere in Abstract.

2) In relation to above, the definition of MACE in this study is not described clearly. Therefore, the focus of this study is blurred.

3) It is difficult to know whether hazard ratios in Table 2 were adjusted for confounding factors or not. Otherwise, it is difficult to conclude that each factors including AF were independently associated with MACE.

Minor comments

1) In Title, I do not think that the current title shows accurately the contents of this study. I hope it will be revied more simply. In addition, an abbreviation should not be used in Title.

2) In Abstract, descriptions of people-year and people-yea should be people-years.

3) Once atrial fibrillation was abbreviated to AF, use it throughout the manuscript.

4) COVID19 should be COVID-19. In addition, COVID-19 should be spelt out at the first time of use. Since COVID-19 already includes “disease”, “COVID-19 disease” seems inappropriate. MACE also includes “event”, “MACE events” seems wrong.

5) Original terms of NOAC should be non-vitamin K antagonist oral anticoagulant, not new anticoagulant. In addition, antivitamin K is generally described as vitamin K antagonist (VKA).

6) Throughout the manuscript CHA2DS2-VASc score should be described correctly.

7) In Table 1, the decimal places of mean, SD, and percentage should be unified to the first decimal point. In addition, death is not a baseline characteristic. It should be deleted from Table 1.

8) Figure 2 or Kaplan-Meier curves cannot be found anywhere.

9) In Page 7, is this number “1,047 the number of MACE” correct?

10) Although authors mentioned odds ratios in the Methods, results using odds rations cannot be found anywhere.

See above.

Author Response

(The authors gave the same response as above.)

Round 2

Reviewer 2 Report

No further comments 

Minor editing

Reviewer 3 Report

This revised manuscript by Molto-Balado P et al. focused on the association between atrial fibrillation (AF) and major adverse cardiovascular events (MACE). Authors revised the manuscript appropriately according to the reviewer’s comments It appeared better.
I do not have further comment to be resolved.